# Preoperative and Postoperative Kinesiophobia Influences Postoperative Outcome Measures Following Anterior Cruciate Ligament Reconstruction: A Prospective Study

**DOI:** 10.3390/jcm12144858

**Published:** 2023-07-24

**Authors:** Umile Giuseppe Longo, Martina Marino, Giacomo Rizzello, Sergio De Salvatore, Ilaria Piergentili, Vincenzo Denaro

**Affiliations:** 1Fondazione Policlinico Universitario Campus Bio-Medico, Via Alvaro del Portillo, 200, 00128 Rome, Italy; g.rizzello@policlinicocampus.it (G.R.); ilaria.piergentili94@gmail.com (I.P.);; 2Research Unit of Orthopaedic and Trauma Surgery, Department of Medicine and Surgery, Università Campus Bio-Medico di Roma, Via Alvaro del Portillo, 21, 00128 Rome, Italy; martilibia@gmail.com (M.M.); s.desalvatore@unicampus.it (S.D.S.); 3Department of Orthopedics, Children’s Hospital Bambino Gesù, Palidoro, 00165 Rome, Italy

**Keywords:** kinesiophobia, ACL-R, postoperative outcome measures, ACL tears

## Abstract

The anterior cruciate ligament (ACL) is the most injured ligament of the knee, and the treatment of choice is usually ACL reconstruction. Kinesiophobia refers to an irrational and paralyzing fear of movement caused by the feeling of being prone to injury or reinjury. The aim of the present study is to evaluate the relationship between preoperative and postoperative kinesiophobia with postoperative outcomes of ACL-R evaluated through SF-36, ACL-RSI, KOOS, and OKS scores. Included patients all underwent ACL reconstruction. The preoperative TSK-13 questionnaire and six-month postoperative TSK-13, ACL-RSI, SF-36, KOOS, and OKS questionnaires were assessed in included patients. Normal distribution was assessed using the Shapiro–Wilk test. The study included 50 patients who filled out the questionnaires at the 6-month postoperative follow-up. Correlations between preoperative TSK-13 and postoperative outcome measures revealed a low–moderate negative correlation between preoperative TSK-13 and SF-36 PCS at 6-month follow-up. Correlations between postoperative TSK-13 and postoperative outcome measures revealed a high negative correlation between preoperative TSK-13 and ACL-RSI, KOOS Symptoms, KOOS Pain, KOOS ADL, and OKS at 6-month follow-up. Preoperative and postoperative kinesiophobia were found to influence postoperative ACL-R outcomes negatively, more specifically an increase in kinesiophobia showed a statistically significant correlation with worse postoperative SF-36 PCS scores in patients.

## 1. Introduction

The anterior cruciate ligament (ACL) is considered to be the most injured ligament of the knee, and the treatment of choice when such a condition causes functional instability is usually ACL reconstruction (ACL-R) [1]. A recent population-based study found that the incidence of ACL tears is approximately 68 per 100,000 person-years, making it a common orthopaedic injury; in the US alone, approximately 100,000 reconstructions are performed annually [2,3,4,5]. Furthermore, large registry databases along with several cohort studies have revealed that the incidence of ACL-R is increasing worldwide [1]. 

In general, musculoskeletal (MSK) conditions such as ACL injury are usually approached using traditional educational models which heavily incorporate the biomechanics, anatomy, and pathoanatomic features of injury [6,7,8]. Such approaches are essential in the medical evaluation of conditions and in determining an appropriate treatment plan; nonetheless, they fail to incorporate the psychosocial component often associated with MSK disorders [6,9,10]. Chronic musculoskeletal pain (CMP) is defined as an ongoing pain felt in joints, bone, and tissues persisting longer than 3 months, and in Western society impacts about 20% of adults [11]. Widely accepted contributors to this kind of pain and its consequent disability include behaviours such as pain catastrophizing, anxiety, fear during movement, and nervous system sensitization [9,11]. These behaviours are at the basis of the development of kinesiophobia in patients suffering from musculoskeletal disorders such as ACL injury. Kinesiophobia is defined as an irrational and paralyzing fear of movement caused by a feeling of being prone to injury or reinjury, and as mentioned is both influenced by and influences chronic pain conditions and associated behaviours [12,13,14]. 

Kinesiophobia is evaluated using the Tampa Scale of Kinesiophobia (TSK) [12]. This condition may be a cause as well as a symptom of disability in orthopaedic patients, and bears a strong psychosocial component [12]. It has two main recognized causes including via a direct painful or traumatic experience or via learned social behaviours [12,13]. Its prevalence in patients suffering from persistent pain usually ranges between 50% and 70% and has been shown to alter the biomechanics of movements and the perception of pain [12]. 

Most recently, studies have shown that biopsychosocial treatment targeting the physical, psychological, and social factors surrounding pain and disability are most effective in the treatment of chronic pain [9,11]. Such findings, as discussed, bring to light the relationship between the psychosocial components of pain, chronic pain conditions, and fear behaviours (including kinesiophobia) and MSK disorders and treatment outcomes. Studying these relationships in practice has the potential to provide lasting change to the approach to treatment in the field of orthopaedics for the benefit of patients. Thus, evaluating these parameters in the context of specific surgical procedures, such as ACL-R, is a step in the right direction. The aim of the present study is to evaluate the relationship between preoperative and postoperative kinesiophobia with postoperative outcomes of ACL-R evaluated through SF-36, ACL-RSI, KOOS, and OKS scores.

## 2. Materials and Methods

In total, 50 patients (19 females and 31 males) who underwent ACL-R were included in the present study. All included patients were operated on by the same expert surgeon at our centre over the course of three years and followed the same standardized rehabilitation protocol. 

The preoperative TSK-13 questionnaire and six-month postoperative TSK-13, ACL-RSI, SF-36, KOOS, and OKS questionnaires were assessed in included patients. Preoperative questionnaires were administered “in-person” by trained nursing staff on the day of intervention. The same nursing staff administered the postoperative questionnaires in the hospital during the 6-month follow-up visit, or alternatively contacted patients via phone. Nurses received training for the administration of the questionnaires and were available to patients for clarification when necessary. Patients were not made aware of the goal of the study when filling the questionnaires.

Six-month follow-up for administration of postoperative questionnaires allows for a sufficient recovery period from the time of intervention to assess meaningful outcome measures. Furthermore, this timeframe allows for the evaluation of the very first changes in quality of life and function following recovery.

### 2.1. Tampa Scale for Kinesiophobia (TSK)

To determine if an individual suffers from kinesiophobia, the Tampa Scale for Kinesiophobia, a patient-reported outcome (PRO) assessment, was used. The TSK contains 17 pieces in its original form; however, there is also a 13-item version (TSK-13) that has superior psychometric qualities, hence its employment in the current investigation [15,16]. The validated Italian language version was used [17]. The possible scores range from 13 (no kinesiophobia) to 52 (maximum kinesiophobia). Neblett et al. defined four severity ranges of kinesiophobia encompassing “subclinical” (13–22), “mild” (23–32), “moderate” (33–42), and “severe” (43–52). In the present study, “No kinesiophobia” is synonymous with subclinical and mild groups, while “Kinesiophobia” with moderate and severe groups [15].

### 2.2. Postoperative Scores

TSK-13, ACL-RSI, SF-36, KOOS, and OKS questionnaires were completed by patients 6 months postoperatively, and for each the validated Italian language version was utilized [18,19,20,21]. Raw values of each subscale were translated into a 0 to 100 scale, going from worst (0) to best (100) condition, to facilitate comparison.

#### 2.2.1. Anterior Cruciate Ligament—Return to Sport after Injury

The ACL-RSI is a questionnaire that evaluates return to sports after injury [18]. It consists of 12 questions, each scored on an 11-point Likert scale from 0 to 100. The total score is determined by summing the scores for each question and expressing them as a percentage, with a range of 0 to 100 indicating the worst and best conditions, respectively.

#### 2.2.2. Short-Form Health Survey-36

The SF-36 questionnaire evaluates general health indicators and generates a corresponding score [19]. It consists of 36 questions divided into eight sections comprising a Physical Component Summary (SF-36 PCS) and a Mental Component Summary (SF-36 MCS), in addition to the inclusion of a single unscaled question regarding health changes in the past year (SF-36 Health Change) [19]. 

#### 2.2.3. Knee Injury and Osteoarthritis Outcome Score

The KOOS is self-administered questionnaire and contains 42 items sectioned into five subscales: Pain, Symptoms, Activities of Daily Living (ADL), Sport and Recreation (Sport/Rec), and Knee-related Quality of Life (QOL). A five-point Likert scale ranging from 0 (no difficulties) to 4 (severe problems) is used to score each question. The final score is calculated by summing the score obtained for each question.

#### 2.2.4. Oxford Knee Score

The OKS (Oxford Knee Score) is a questionnaire consisting of 12 items that evaluate pain and daily living activities related to the knee. The questionnaire is divided into two subscales for pain and function, each comprising five response categories that are scored on a 1–5 Likert scale. One may obtain a minimum total score of 12 and a maximum total score of 60. 

### 2.3. Statistical Analysis

A priori power analysis was performed, with a level of significance of 0.05, a statistical power of 80%, a correlation of −0.48 between TSK and ACL-RSI, and a minimum total sample size amounting to 31 subjects [18].

To evaluate the normal distribution of the data, the Shapiro–Wilk test was employed. As the data did not exhibit a normal distribution, the Independent-Samples Mann–Whitney U Test was used to calculate the variations in scores between the groups (preoperative kinesiophobia versus no preoperative kinesiophobia and postoperative kinesiophobia versus no postoperative kinesiophobia). Additionally, the relationship between preoperative TSK-13 scores and postoperative scores was measured via Spearman’s correlation.

SPSS for Windows (version 26; Armonk, NY: IBM Corp), R Core Team (2020) version 4.0.3, and SAS OnDemand for Academics were used to perform all statistical analyses and produce figures.

## 3. Results

Overall, 50 eligible patients (19 females and 31 males) were included in the study. All patients completed the questionnaires at the 6-month postoperative follow-up. All included patients are of Italian nationality. The mean age at the time of intervention was 39 in females, 30.1 in males, and 33.6 overall. The oldest included patient was a 63-year-old female and the youngest was a 16-year-old male.

### 3.1. Preoperative Kinesiophobia

At preoperative follow-up, 32 patients (64%) were classified in the “kinesiophobia” group, whereas 18 were classified in the “no kinesiophobia” group (36%). 

The correlation between preoperative TSK-13 and postoperative outcome measures revealed a low–moderate negative correlation between preoperative TSK-13 and SF-36 PCS (rho = −0.292, *p* = 0.04) at the 6-month follow-up, Table 1. 

As the preoperative TSK-13 score increases, the SF-36 PCS decreases.

Statistically significant differences between the two groups (preoperative “kinesiophobia” vs. preoperative “no kinesiophobia”) were found in postoperative SF-36 PCS (*p* = 0.008), Figure 1.

Patients without preoperative kinesiophobia showed higher values of the SF-36 PCS (Table 2).

### 3.2. Postoperative Kinesiophobia

At the postoperative follow-up, 21 patients (42%) were classified in the “kinesiophobia” group, whereas 29 were classified in the “no kinesiophobia” group (58%). 

The correlation between postoperative TSK-13 and postoperative outcome measures revealed a high negative correlation between preoperative TSK-13 and ACL-RSI (rho = −0.486, *p* < 0.001), KOOS Symptoms (rho = −0.513, *p* < 0.001), KOOS Pain (rho= −0.565, *p* < 0.001), KOOS ADL (rho = −0.496, *p* < 0.001), and OKS (rho = −0.703, *p* < 0.001) at the 6-month follow-up.

As the preoperative TSK-13 score increases, ACL-RSI, KOOS Symptoms, KOOS Pain, KOOS ADL, and OKS decrease.

Statistically significant differences between the two groups (postoperative “kinesiophobia” vs. postoperative “no kinesiophobia”) were found in postoperative ACL-RSI (*p* < 0.001), KOOS Symptoms (*p* < 0.001), KOOS Pain (*p* < 0.001), KOOS ADL (*p* < 0.001), and OKS (*p* < 0.001). All the results are summarized in Figure 2, Figure 3 and Figure 4.

Patients without preoperative kinesiophobia showed higher values in ACL-RSI, KOOS Symptoms, KOOS Pain, KOOS ADL and OKS (Table 3).

## 4. Discussion

The results suggest that patients suffering from kinesiophobia preoperatively and postoperatively suffer from worse ACL-R outcomes as measured via the utilized scores. Specifically, a negative correlation was found between an increase in TSK-13 score preoperatively and postoperative SF-36 PCS score, and between an increase in TSK-13 score postoperatively and postoperative ACL-RSI, KOOS Symptoms, KOOS Pain, KOOS ADL, and OKS scores.

A low–moderate negative correlation was identified between preoperative TSK-13 and postoperative SF-36 PCS at the 6-month follow-up. As the TSK-13 score increased in the “kinesiophobia” group, the postoperative SF-36 PCS score decreased. Conversely, patients placed in the “no kinesiophobia” group showed statistically significantly higher SF-36 PCS values postoperatively. The PCS assesses physical function, role limitations caused by physical problems, bodily pain, and general health across four predetermined domains. A lower score on this assessment indicates poorer overall physical health, and in this case, the decline in the SF-36 PCS value appears to be linked to kinesiophobia. Brown et al.’s systematic review evaluated the influence of kinesiophobia on outcomes after total knee arthroplasty (TKA), concluding that its presence negatively impacted functional outcomes for up to a year after surgery [22]. This is consistent with the present findings on ACL-R. Furthermore, research has shown that kinesiophobia can independently predict postsurgical function, even when controlling for other psychological and physical variables [22].

The term kinesiophobia inherently encompasses a multitude of behaviours and attitudes regarding pain and movement, including pain catastrophizing, fear of movement, and possibly associated mood disorders such as anxiety and depression [23]. Such attitudes towards pain have been recognized in some cases as predictors of poor recovery from TKA and several studies show that they may be some of the strongest psychological predictors of return to work following orthopaedic injury [23,24]. The postoperative SF-36 PCS score itself may serve as a tool to measure a patient’s ability to return to daily activities, work, and perhaps even sports thanks to its quantitative evaluation of physical well-being. Thus, kinesiophobia experienced before ACL-R may be at play in patients’ ability or inability to return to their full physical potential following surgery given its negative effects on surgical outcomes as measured via the SF-36 PCS score.

A high negative correlation was found between postoperative TSK-13 and postoperative outcome measures; in fact, as the preoperative TSK-13 score increased ACL-RSI, KOOS Symptoms, KOOS Pain, KOOS ADL, and OKS values were found to decrease. Furthermore, patients without preoperative kinesiophobia in the first place showed statistically significant higher values for ACL-RSI, KOOS Symptoms, KOOS Pain, KOOS ADL, and OKS.

An evaluation of the occurrence of postoperative fear of movement in TKA and its effects on outcome measures postoperatively found that the OKS score at 6-month follow-up was significantly lower in patients who suffered from fear of movement [25]. Similarly, in the present study, it was discovered that patients with postoperative kinesiophobia following ACL-R showed lower OKS scores compared to their “no kinesiophobia” counterparts. Although TKA and ACL-R are different surgical procedures in both cases, there seems to be a negative relationship between postoperative kinesiophobia and specific surgical outcomes.

Return to sports after injury, measured via ACL-RSI score, was found to decrease with an increase in kinesiophobia, indicating that patients affected by this condition are less likely to return to physical activity. This score is measured based on three domains—emotions, confidence, and risk appraisal—ultimately evaluating emotional and physiological readiness to return to sports (RTS) following injury and surgical intervention. The extent of self-reported fear in some studies was shown to predict both second injury to ACL and return to sport [26]. Patients who reported higher levels of fear during the RTS phase of recovery were more likely to exhibit reduced activity levels, experience asymmetry during the single leg hop for distance assessment, and have an elevated risk of experiencing a second ACL injury within 24 months [26]. These results are in line with the present findings given that at the base of kinesiophobia is the fear of movement. Another study even found that kinesiophobia negatively affects range of motion (ROM) and that this issue is not resolved by simply demonstrating patients’ ROM in theatre [22]. The exclusion of patient-reported fear of movement from discharge planning algorithms following ACL-R is commonplace and its introduction as an evaluating factor and possibly its integration into rehabilitation programs could yield positive outcomes for patients [26]. Rehabilitation programs alone are crucial for the achievement of successful postoperative outcomes and as a result should also tend to the psychosocial components of recovery [27]. Overall, concordant findings stress the need to incorporate strategies to tackle preoperative and postoperative kinesiophobia to promote RTS and return to daily activity following ACL-R. 

The results showed that KOOS Symptoms, Pain, and ADL domains were all worsened in patients suffering from postoperative kinesiophobia. It is likely that kinesiophobia and the worsening of these outcome measures have a mutual influence on one another. It has been reported, for example, that negative outcome expectations (to which kinesiophobia often contributes) are detrimental to patient motivation, reducing the likelihood of individuals to initiate the behaviours necessary to resume ADL and sports following certain orthopaedic procedures [23]. Knowing the positive direct and indirect effects that physical activity has on mental health, an increasingly sedentary lifestyle may aggravate an already precarious psychological condition, possibly worsening the kinesiophobia that is already at play in affected patients whilst hindering recovery [28]. Such observations in combination with the present findings further stress the need for consideration and awareness of this condition in the field of orthopaedics, especially in postoperative rehabilitation programs, to, amongst other things, maximize patients’ ability to RTS, return to ADL, and allow the achievement of full ROM postoperatively.

A pilot study by Coronado et al. evaluated the role of psychosocial states and behaviours on postsurgical outcomes by trialling a telephone-delivered cognitive-behavioural-based intervention to be implemented prior to and immediately after ACL-R [3]. It was designed to target a variety of psychological factors including those influencing fear behaviours, which are intrinsic to kinesiophobia [3]. Following this trial, meaningful reductions in fear of movement or reinjury and catastrophizing and an increase in self-efficacy were observed, and authors even reported an increase in KOOS QOL 6 months postoperatively [3]. The combination of conducting further research regarding the influence of kinesiophobia on orthopaedic procedures and the development of introspective therapeutic programs incorporating approaches oriented towards targeting psychosocial factors is a step in the right direction. The development of novel additions to rehabilitation programs perhaps has the potential to be life-changing for many orthopaedic patients by improving postoperative outcomes.

### Limitations

The limitations of the present study include the lack of randomization, which inevitably increases the likelihood of confounding factors and bias. Furthermore, the use of patient-reported outcome measures (PROMs) may influence the comparability of results across samples given the introduction of patient subjectivity. To minimize this, data were collected utilizing well-researched surveys, translated professionally into the Italian language, and the questionnaire procedure was carefully explained to the included patients. In addition to this, surveys were performed 6-months postoperatively, allowing a substantial recovery period and making the results more comparable. 

In addition to this, more data regarding patient demographics and a subsequent stratification of results may be beneficial in future studies. Stratifying patients by characteristics such as sex and age may provide more information on the influence of kinesiophobia in specific subgroups.

Furthermore, much of the literature referenced regarding the relationships between kinesiophobia and postoperative outcomes regarded TKA instead of ACL-R; such data were utilized nonetheless as a means of comparison in the present study. Regardless, the lack of available literature on the matter highlights the necessity for more research in this field. 

## 5. Conclusions

Both preoperative and postoperative kinesiophobia were found to influence postoperative ACL-R outcomes negatively. In the case of preoperative kinesiophobia, SF-36 PCS was found to have a negative relationship with increasing kinesiophobia. In postoperative kinesiophobia, a negative relationship was recognized between increasing kinesiophobia and ACL-RSI, SF-36, KOOS, and OKS scores. Psychosocial factors have been continuously shown to influence overall health and the present study attests to that. Patients suffering from this complex cognitive–behavioural condition closely associated with pain and fear were found to suffer from worse ACL-R outcomes, which possibly contributed to a slowed or impeded recovery. Ultimately, more studies are needed regarding the role of psychosocial factors, such as kinesiophobia, on postoperative outcomes in the field of orthopaedic surgery, as well as more effort towards the development of preoperative and postoperative therapeutic and rehabilitation programs that tend to these conditions. 

## Figures and Tables

**Figure 1 jcm-12-04858-f001:**
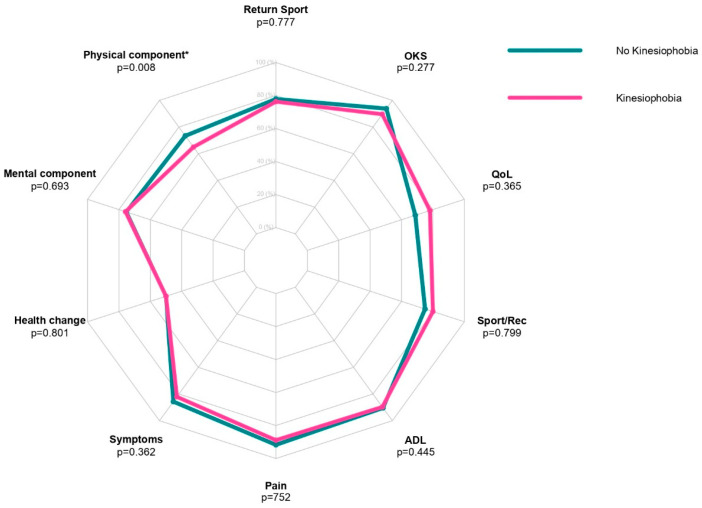
Radar plot of preoperative kinesiophobia and no kinesiophobia patients assessing RTS, OKS, KOOS QoL, KOOS Sport/Rec, KOOS ADL, KOOS Pain, KOOS Symptoms, SF-36 Health Change, SF-36 MCS, and SF-36 PCS scores. * = *p* < 0.05.

**Figure 2 jcm-12-04858-f002:**
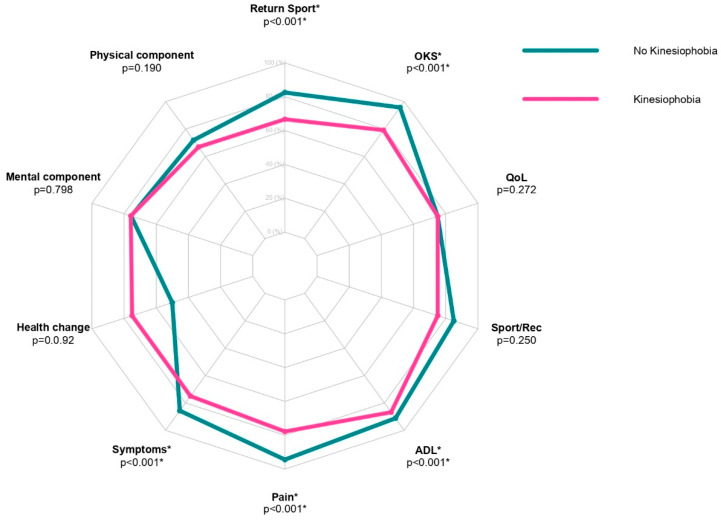
Radar plot of postoperative kinesiophobia and no kinesiophobia patients assessing SRTS, OKS, KOOS QoL, KOOS Sport/Rec, KOOS ADL, KOOS Pain, KOOS Symptoms, SF-36 Health Change, SF-36 MCS, and SF-36 PCS scores. * = *p* < 0.05.

**Figure 3 jcm-12-04858-f003:**
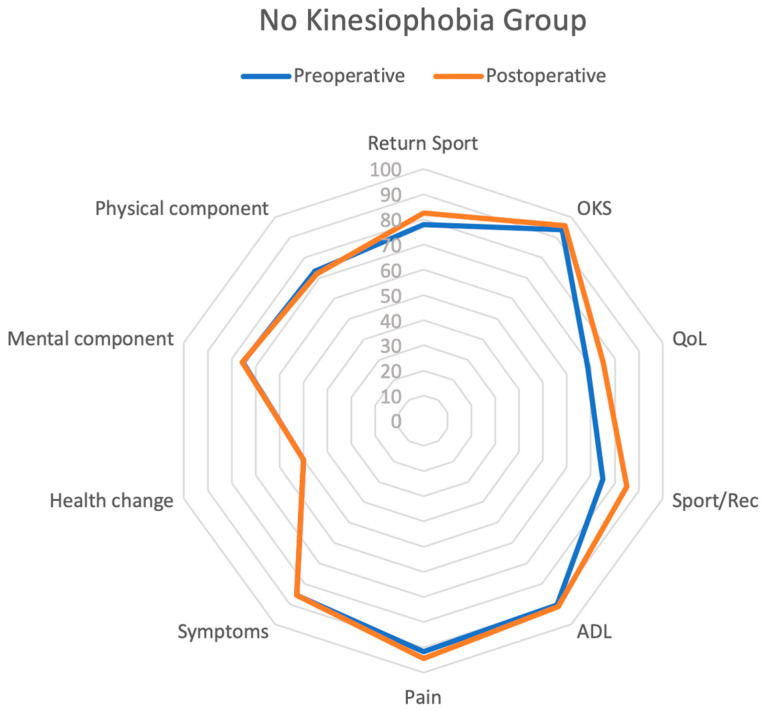
Radar plot of preoperative and postoperative outcomes in no kinesiophobia group assessing RTS, OKS, KOOS QoL, KOOS Sport/Rec, KOOS ADL, KOOS Pain, KOOS Symptoms, SF-36 Health Change, SF-36 MCS, and SF-36 PCS scores.

**Figure 4 jcm-12-04858-f004:**
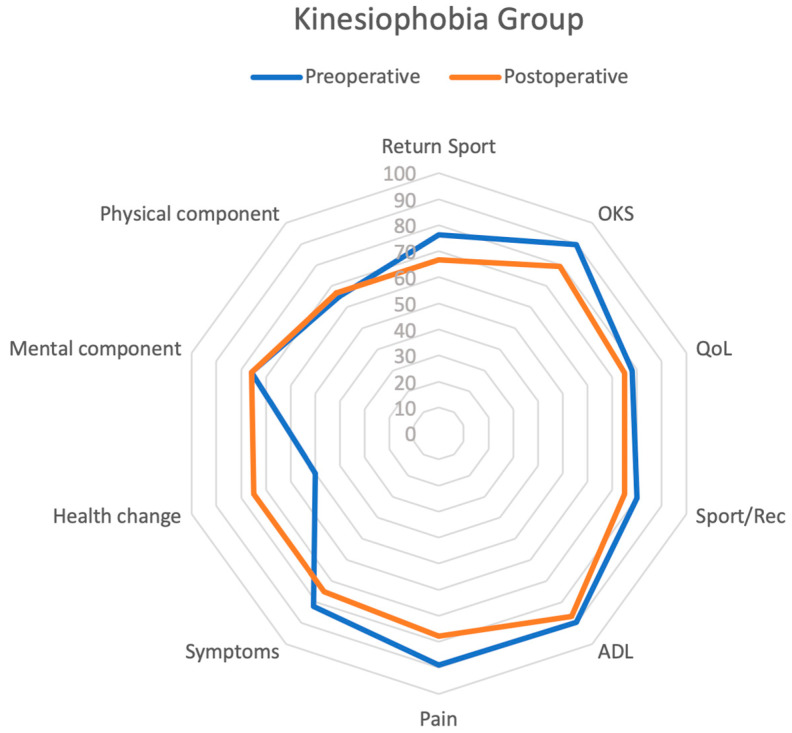
Radar plot of preoperative and postoperative outcomes in kinesiophobia group assessing RTS, OKS, KOOS QoL, KOOS Sport/Rec, KOOS ADL, KOOS Pain, KOOS Symptoms, SF-36 Health Change, SF-36 MCS, and SF-36 PCS scores.

**Table 1 jcm-12-04858-t001:** Correlations between preoperative TSK-13 and postoperative scores and correlations between postoperative TSK-13 and postoperative scores.

Parameter	Correlation between Preoperative TSK-13 and Postoperative Scores		Correlation between Postoperative TSK-13 and Postoperative Scores	
	rho	*p*-Value	rho	*p*-Value
ACL-RSI	−0.013	0.926	−0.486	<0.001 *
SF-36 PCS	−0.292	0.040 *	−0.248	0.083
SF-36 MCS	−0.094	0.516	−0.095	0.512
SF-36 Health change	−0.111	0.441	0.227	0.113
KOOS Symptoms	−0.041	0.776	−0.513	<0.001 *
KOOS Pain	0.025	0.866	−0.565	<0.001 *
KOOS ADL	−0.040	0.785	−0.496	<0.001 *
KOOS Sport/Rec	0.051	0.726	−0.274	0.054
KOOS QoL	0.121	0.401	−0.186	0.196
OKS	−0.107	0.459	−0.703	<0.001 *

TSK-13: Tampa Scale for Kinesiophobia; ACL-RSI: Anterior Cruciate Ligament—Return to Sport After Injury; SF-36 PCS: 36-item Short-Form Health Survey—Physical Component Summary; MCS: Mental Component Summary; KOOS ADL: Knee Injury and Osteoarthritis Outcome Score—Activities of Daily Living; Sport/Rec: Sport and Recreation; QoL: Quality of Life; OKS: Oxford Knee Score. * *p* < 0.05.

**Table 2 jcm-12-04858-t002:** Median (min–max) values of postoperative scores compared to preoperative kinesiophobia.

Parameter	No Kinesiophobia (N = 18)		Kinesiophobia (N = 32)		*p*-Value
	Median	Range	Median	Range	
ACL-RSI	77.9	52.5–100	76.3	42.5–96.7	0.777
SF-36 PCS	73.4	45–91.3	65.0	29.4–87.5	0.008 *
SF-36 MCS	75.3	57–97.5	75.8	48–93.3	0.693
SF-36 Health change	50.0	25–100	50.0	25–100	0.801
KOOS Symptoms	85.7	60.7–100	82.1	50–100	0.362
KOOS Pain	91.7	66.7–100	88.9	52.8–100	0.752
KOOS ADL	90.4	61.8–100	89.7	51.5–98.5	0.445
KOOS Sport/Rec	75.0	45–100	80.0	20–100	0.799
KOOS QoL	68.8	31.3–100	78.1	12.5–100	0.365
OKS	93.8	45.8–100	89.6	41.7–100	0.277

ACL-RSI: Anterior Cruciate Ligament—Return to Sport After Injury; SF-36 PCS: 36-item Short-Form Health Survey—Physical Component Summary; MCS: Mental Component Summary; KOOS ADL: Knee Injury and Osteoarthritis Outcome Score—Activities of Daily Living; Sport/Rec: Sport and Recreation; QoL: Quality of Life; OKS: Oxford Knee Score. * *p* < 0.05

**Table 3 jcm-12-04858-t003:** Median (min–max) values of postoperative scores compared to postoperative Kinesiophobia.

Parameter	No Kinesiophobia (N = 29)		Kinesiophobia (N = 21)		*p*-Value
	Median	Range	Median	Range	
ACL-RSI	82.5	63.3–100	66.7	42.5–88.3	<0.001 *
SF-36 PCS	71.9	38.8–91.3	66.9	29.4–91.3	0.190
SF-36 MCS	75.6	57–97.5	75.8	48–97.5	0.798
SF-36 Health change	50.0	25–100	75.0	25–100	0.092
KOOS Symptoms	85.7	78.6–100	75.0	50–92.9	<0.001 *
KOOS Pain	94.4	77.8–100	77.8	52.8–94.4	<0.001*
KOOS ADL	91.2	77.9–100	86.8	51.5–98.5	<0.001 *
KOOS Sport/Rec	85.0	50–100	75.0	20–100	0.250
KOOS QoL	75.0	37.5–100	75.0	12.5–100	0.272
OKS	95.8	81.3–100	79.2	41.7–95.8	<0.001 *

ACL-RSI: Anterior Cruciate Ligament—Return to Sport After Injury; SF-36 PCS: 36-item Short-Form Health Survey—Physical Component Summary; MCS: Mental Component Summary; KOOS ADL: Knee Injury and Osteoarthritis Outcome Score—Activities of Daily Living; Sport/Rec: Sport and Recreation; QoL: Quality of Life; OKS: Oxford Knee Score. * *p* < 0.05

## Data Availability

The datasets used and/or analysed during the current study are available from the corresponding author upon reasonable request.

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
