# Peer review of "Preoperative and Postoperative Kinesiophobia Influences Postoperative Outcome Measures Following Anterior Cruciate Ligament Reconstruction: A Prospective Study"

_jcm, 2023, doi:10.3390/jcm12144858_

Round 1

Reviewer 1 Report

Dear authors,

I am pleased to review the submitted paper JCM-2439294 entitled "Preoperative and Postoperative Kinesiophobia Influences Post-operative Outcome Measures Following ACL Reconstruction: A Prospective Study"

The present paper focuses on evaluate the relationship between preoperative and postoperative kinesiophobia with postoperative outcomes of ACL-R evaluated through SF-36, ACL-RSI, SF-36, KOOS and OKS scores.

In my opinion the content is original, current, but not objective and persuasive.

1. Results:”Overall, 50 eligible patients (19 females and 31 males) were included in the study.” The demographic characteristics of this study is missing including age, nationality...

2. Methods: “The preoperative TSK-13 questionnaire and six-month postoperative TSK-13, ACL-RSI, SF-36, KOOS and OKS questionnaires were assessed in included patients.”The detailed questionnaire obtain methods is critical, are the patients finish the questionnaire totally by themselves or with help of others? How many medical stuff were involved in this survey?

3. Methods: “The preoperative TSK-13 questionnaire and six-month postoperative TSK-13, ACL-RSI, SF-36, KOOS and OKS questionnaires were assessed in included patients.” Why the authors choose timeframe preoperative for 6 months? Did patients follow the same rehabilitation program?

4. Table: This study entitled "Preoperative and Postoperative Kinesiophobia Influences Post-operative Outcome Measures Following ACL Reconstruction: A Prospective Study", table should included Preoperative and Postoperative in the same table for compare.

5. Figure: This study entitled "Preoperative and Postoperative Kinesiophobia Influences Post-operative Outcome Measures Following ACL Reconstruction: A Prospective Study", figure should included Preoperative and Postoperative in the same figure for compare.

Minor editing of English language required.

Author Response

REVIWER 1

Dear authors,

I am pleased to review the submitted paper JCM-2439294 entitled "Preoperative and Postoperative Kinesiophobia Influences Post-operative Outcome Measures Following ACL Reconstruction: A Prospective Study". The present paper focuses on evaluate the relationship between preoperative and postoperative kinesiophobia with postoperative outcomes of ACL-R evaluated through SF-36, ACL-RSI, SF-36, KOOS and OKS scores. In my opinion the content is original, current, but not objective and persuasive.

            Thank you very much for the comments provided, we carefully read through each and resolved them to the best of our abilities. Our materials and methods section benefited the most from your review and we believe we have strongly improved it. Our tables also benefitted from some rearranging and are now presenting the results more clearly.

  1. Results: “Overall, 50 eligible patients (19 females and 31 males) were included in the study.” The demographic characteristic of this study is missing including age, nationality...

            Thank you for this comment, unfortunately this data is not retrievable and as a result it wasn’t possible to add it to the study, a comment on this matter was added in the limitations section (line 341-344).

  1. Methods: “The preoperative TSK-13 questionnaire and six-month postoperative TSK-13, ACL-RSI, SF-36, KOOS and OKS questionnaires were assessed in included patients.” The detailed questionnaire obtain methods is critical, are the patients finish the questionnaire totally by themselves or with help of others? How many medical stuff were involved in this survey?

            We didn’t think to add a more thorough explanation of this procedure in the materials and methods section, however, we agree that it is a very important part of the study design. Thus, we added some sentences with the answers to each of these points in the methods of the study (lines 81-86).

  1. Methods: “The preoperative TSK-13 questionnaire and six-month postoperative TSK-13, ACL-RSI, SF-36, KOOS and OKS questionnaires were assessed in included patients.” Why the authors choose timeframe preoperative for 6 months? Did patients follow the same rehabilitation program?

            Thank you for this comment, once again we agree that this information is crucial to add in the methods of our study and we did include the answers to these questions on lines 88-91.

  1. Table: This study entitled "Preoperative and Postoperative Kinesiophobia Influences Post-operative Outcome Measures Following ACL Reconst ruction: A Prospective Study", table should included Preoperative and Postoperative in the same table for compare.

We appreciate your comment and in order to resolve this issue we merged table 1 and table 3 in order to include both correlations between preoperative TSK-13 and postoperative scores and correlations between postoperative TSK-13 and postoperative scores in the same table for an easier comparaison.

  1. Figure: This study entitled "Preoperative and Postoperative Kinesiophobia Influences Post-operative Outcome Measures Following ACL Reconstruction: A Prospective Study", figure should included Preoperative and Postoperative in the same figure for compare.

We appreciate your comment however the figures included illustrate the comparison between preoperative and postoperative scores in “kinesiophobia” and “no kinesiophobia” groups separately because we are focusing on the differences between groups more so than the preoperative and postoperative change. As a result, the figures were not changed.

English language review was carried out by a native-English speaker.

Reviewer 2 Report

Abstract

1. Mentions the use of "SF-36" twice in a single sentence (line 19), which might confuse readers. It's important to mention each measurement tool only once or clarify if there are different components of the SF-36 being used.
2. The abstract could be more explicit about the negative correlation between preoperative kinesiophobia and postoperative outcomes. The current phrasing (line 29-30) is a bit convoluted

Introduction

1. Explanation about kinesiophobia (line 50-57) seems suddenly introduced after a section on chronic musculoskeletal pain (line 45-49). Kinesiophobia could be introduced as a specific case of behaviors that contribute to chronic pain, ensuring a smoother transition between ideas.

2. SF-36 outcome measure is mentioned twice in the last line (line 67-68).

3. Citation at the end of the line about musculoskeletal conditions (line 44) could be checked to make sure it covers the full range of ideas presented in the paragraph. Furthermore, ensure that your text doesn't heavily rely on one source, such as [13], which is mentioned repeatedly.

Methods

1. introduction of the patients (line 70) could benefit from including more details such as age range, the criteria for their selection, and if there was a control group.

2. It would be helpful to add more information about when and how the questionnaires were administered. Did the same individual conduct all the assessments? Were the patients aware of the goal of the study?

3. The explanation of how the scores were calculated for different questionnaires (line 88-106) could be more consistent. For instance, you might explain all scoring methods as ranges from worst to best conditions, as you've done for the ACL-RSI, SF-36, and KOOS questionnaires.

4. Each questionnaire could have its own subsection or paragraph, allowing for a more segmented, clear, and readable presentation of each tool used in the study. This would also enable readers to easily find the information they are looking for.

Results

They are well presented in Tables and Figures. However, some figures font is too small and hard to read. Please revise the size of the text.

Discussions

They are well written, presenting both advantages and limitations according to the current state of the art.

Conclusion

Ok

References

Ok and recent cited papers.

English language and spelling is normal.

Author Response

REVIEWR 2

We thank you for the comments that were provided which enabled us to greatly improve our study. We believe that by resolving your comments we have improved the readability of our manuscript, as well as included important citations in our introduction and important information in the methods sections that were missing before-hand.

  1. Abstract: Mentions the use of "SF-36" twice in a single sentence (line 19), which might confuse readers. It's important to mention each measurement tool only once or clarify if there are different components of the SF-36 being used.

Thank you very much for pointing out this typo, the second ‘SF-36’ present in the text was deleted.

  1. Abstract: The abstract could be more explicit about the negative correlation between preoperative kinesiophobia and postoperative outcomes. The current phrasing (line 29-30) is a bit convoluted

            We agree that the wording of that sentence was confusing and as a result re-phrased this concept as seen on lines 29-30.

  1. 3. Introduction:Explanation about kinesiophobia (line 50-57) seems suddenly introduced after a section on chronic musculoskeletal pain (line 45-49). Kinesiophobia could be introduced as a specific case of behaviours that contribute to chronic pain, ensuring a smoother transition between ideas.

            Thank you for this comment, to make this transition smoother we re-worded and re-organized the text on lines 54-58. We believe that these changes have improved the quality of our introduction.

  1. 4. Introduction: SF-36 outcome measure is mentioned twice in the last line (line 67-68).

Thank you once again for identifying this mistake, the second ‘SF-36’ was deleted from the text.

  1. 5. Introduction: Citation at the end of the line about musculoskeletal conditions (line 44) could be checked to make sure it covers the full range of ideas presented in the paragraph. Furthermore, ensure that your text doesn't heavily rely on one source, such as [13], which is mentioned repeatedly.

            We agree that more citations were necessary to support the claims on line 44, in fact Healey et. al. was added to the citations and is a study that looks over intervention development for patients suffering from musculoskeletal pain. Furthermore, to enrich the bibliography and to find more support to the claims made in the introduction two other concordant studies were included (Rodrigo Nuñez-Cortes et. al. and Pontillo et. al.). All three cited studies were published in 2023 and support the information presented in the text.

  1. Methods: introduction of the patients (line 70) could benefit from including more details such as age range, the criteria for their selection, and if there was a control group.

Thank you for your comment. Unfortunately, we were unable to retrieve this data, so it was not included in the study. We have added a comment on this matter in the limitations section which can be seen on lines 341-344.

  1. Methods: It would be helpful to add more information about when and how the questionnaires were administered. Did the same individual conduct all the assessments? Were the patients aware of the goal of the study?

Thank you very much for this comment, we agree that we did not provide a comprehensive explanation of this procedure in the materials and methods section, despite its importance to the study design. Therefore, we have added some sentences to the methods section that address each of these points (lines 81-86).

  1. Methods: The explanation of how the scores were calculated for different questionnaires (line 88-106) could be more consistent. For instance, you might explain all scoring methods as ranges from worst to best conditions, as you've done for the ACL-RSI, SF-36, and KOOS questionnaires.

            Thank you for your comment, in order to solve this issue we separated the description of each questionnaire using separate headings and added a sentence which explained the standardization of scores which can be seen on lines 125-126.

  1. Methods: Each questionnaire could have its own subsection or paragraph, allowing for a more segmented, clear, and readable presentation of each tool used in the study. This would also enable readers to easily find the information they are looking for.

            We think that separating the description of postoperative questionnaires using sub-headings made our text clearer and as a result we thank you very much for this suggestion.

  1. Results: They are well presented in Tables and Figures. However, some figures font is too small and hard to read. Please revise the size of the text.

We agree that the figures were presented with a text that was too small to read and in order to fix this issue we cropped the images and enlarged them.

Round 2

Reviewer 1 Report

Dear authors,

I am pleased to review the revised paper JCM-2439294 entitled "Preoperative and Postoperative Kinesiophobia Influences Post-operative Outcome Measures Following ACL Reconstruction: A Prospective Study". The present paper focuses on evaluate the relationship between preoperative and postoperative kinesiophobia with postoperative outcomes of ACL-R evaluated through SF-36, ACL-RSI, SF-36, KOOS and OKS scores. In my opinion the content is original, current, but not objective and persuasive.

I reviewed this paper before and the authors have now submitted a revised version. I read the authors' response to the reviewer comments and some of the questions has been answered but some of them are still be unsolved.

1.Results: “Overall, 50 eligible patients (19 females and 31 males) were included in the study.” The demographic characteristic of this study is missing including age, nationality...

A:Thank you for this comment, unfortunately this data is not retrievable and as a result it wasn’t possible to add it to the study, a comment on this matter was added in the limitations section (line 341-344).

Question " The demographic characteristic of this study is missing including age, nationality..." unsolved.

2.Methods: “The preoperative TSK-13 questionnaire and six-month postoperative TSK-13, ACL-RSI, SF-36, KOOS and OKS questionnaires were assessed in included patients.” The detailed questionnaire obtain methods is critical, are the patients finish the questionnaire totally by themselves or with help of others? How many medical stuff were involved in this survey?

A:We didn’t think to add a more thorough explanation of this procedure in the materials and methods section, however, we agree that it is a very important part of the study design. Thus, we added some sentences with the answers to each of these points in the methods of the study (lines 81-86).

Question solved . 

3.Methods: “The preoperative TSK-13 questionnaire and six-month postoperative TSK-13, ACL-RSI, SF-36, KOOS and OKS questionnaires were assessed in included patients.” Why the authors choose timeframe preoperative for 6 months? Did patients follow the same rehabilitation program?

A:Thank you for this comment, once again we agree that this information is crucial to add in the methods of our study and we did include the answers to these questions on lines 88-91.

Question solved . 

Table: This study entitled "Preoperative and Postoperative Kinesiophobia Influences Post-operative Outcome Measures Following ACL Reconst ruction: A Prospective Study", table should included Preoperative and Postoperative in the same table for compare.

We appreciate your comment and in order to resolve this issue we merged table 1 and table 3 in order to include both correlations between preoperative TSK-13 and postoperative scores and correlations between postoperative TSK-13 and postoperative scores in the same table for an easier comparaison.

Question solved .  

Figure: This study entitled "Preoperative and Postoperative Kinesiophobia Influences Post-operative Outcome Measures Following ACL Reconstruction: A Prospective Study", figure should included Preoperative and Postoperative in the same figure for compare.

We appreciate your comment however the figures included illustrate the comparison between preoperative and postoperative scores in “kinesiophobia” and “no kinesiophobia” groups separately because we are focusing on the differences between groups more so than the preoperative and postoperative change. As a result, the figures were not changed.

Question unsolved . 
